# Temporally-Extended Prompts Optimization for SAM in Interactive Medical Image Segmentation

**Chuyun Shen** [1]   **Wenhao Li** [2]   **Ya Zhang** [3]   **Xiangfeng Wang** [1]

## Abstract

The *Segmentation Anything Model* (SAM) has recently emerged as a foundation model for addressing image segmentation. Owing to the intrinsic complexity of medical images and the high annotation cost, the medical image segmentation (MIS) community has been encouraged to investigate SAM's zero-shot capabilities to facilitate automatic annotation. Inspired by the extraordinary accomplishments of *interactive* medical image segmentation (IMIS) paradigm, this paper focuses on assessing the potential of SAM's zero-shot capabilities within the IMIS paradigm to amplify its benefits in the MIS domain. Regrettably, we observe that SAM's vulnerability to prompt forms (e.g., points, bounding boxes) becomes notably pronounced in IMIS. This leads us to develop a mechanism that adaptively offers suitable prompt forms for human experts. We refer to the mechanism above as *temporally-extended prompts optimization* (TEPO) and model it as a Markov decision process, solvable through reinforcement learning. Numerical experiments on the standardized benchmark `Brats2020` demonstrate that the learned TEPO agent can further enhance SAM's zero-shot capability in the MIS context.

## 1. Introduction

The *Segmentation Anything Model* (SAM) (Kirillov et al., 2023) has recently been proposed as a foundational model for addressing image segmentation problems. SAM's effectiveness is principally evaluated in natural image domains, demonstrating a remarkable prompt-based, zero-shot gener-

alization capability. Segmentation within medical images (MIS), on the other hand, presents complex challenges owing to their substantial deviation from natural images, encompassing multifaceted modalities, intricate anatomical structures, indeterminate and sophisticated object boundaries, and extensive object scales (Sharma & Aggarwal, 2010; Hesamian et al., 2019; Huang et al., 2023).

Predominant MIS methods principally employ domain-specific architectures and necessitate reliance upon massive, high-quality expert annotations (Ronneberger et al., 2015; Isensee et al., 2021; Zhou et al., 2021; Cao et al., 2023). In light of the considerable expenditure incurred by dense labeling, the community has embarked on exploring SAM's zero-shot generalization capabilities in MIS tasks, thereby fostering automated annotation of medical images (Ji et al., 2023a;b; Mohapatra et al., 2023; Deng et al., 2023; Zhou et al., 2023; He et al., 2023; Mazurowski et al., 2023; Ma & Wang, 2023; Cheng et al., 2023; Zhang & Jiao, 2023; Roy et al., 2023; Huang et al., 2023; Mattjie et al., 2023).

Motivated by the remarkable achievements of *interactive* medical image segmentation (IMIS), this paper goes a step further and centers on investigating the potential of zero-shot capabilities of SAM in IMIS to magnify the advantages of SAM in MIS domain. Many works demonstrate the significant performance enhancement attributable to the IMIS paradigm (Xu et al., 2016; Rajchl et al., 2016; Lin et al., 2016; Castrejon et al., 2017; Wang et al., 2018; Song et al., 2018; Liao et al., 2020; Ma et al., 2021; Li et al., 2021). Specifically, IMIS overcomes the performance limitation inherent in end-to-end MIS approaches by reconceptualizing MIS as a multi-stage, human-in-the-loop task. At each iteration, medical professionals impart valuable feedback (e.g., designating critical points, demarcating boundaries, or construing bounding boxes) to identify inaccuracies in the model output. Consequently, the model refines the segmentation results following human feedback.

The congruity between the human feedback forms and the prompt forms in SAM facilitates the seamless integration of SAM. Nevertheless, recent investigations reveal that, in contrast to natural image segmentation, the susceptibility of SAM to prompt forms (e.g., points or bounding boxes) is significantly heightened within MIS tasks, resulting in

---

[1]School of Computer Science and Technology, East China Normal University, Shanghai 200062, China [2]School of Data Science, The Chinese University of Hong Kong, Shenzhen, Shenzhen 518172, China [3]Cooperative Medianet Innovation Center, Shanghai Jiao Tong University, Shanghai, 200240, China. Correspondence to: Xiangfeng Wang <xfwang@cs.ecnu.edu.cn>.

*Interactive Learning with Implicit Human Feedback Workshop at ICML 2023.*

substantial discrepancies in zero-shot performance when various prompt forms are employed (Cheng et al., 2023; Roy et al., 2023; Zhang & Jiao, 2023). Regrettably, we find this issue is markedly exacerbated within the IMIS context.

This phenomenon can be attributed to two primary factors. Firstly, the segmentation stages are interdependent; the previous prompt forms selection influences the ensuing segmentation and the choice of subsequent prompt forms. Secondly, human experts display preferences and stochasticity in their feedback, seldom contemplating the ramifications of the prompt forms on the performance and the intricate interconnections between antecedent and successive prompt forms. Consequently, this revelation impels us to recommend the most efficacious prompt forms for human feedback at each successive IMIS stage, a challenge we designate as *temporally-extended prompts optimization*.

As a formidable instrument for addressing sequential decision-making, reinforcement learning (RL) (Sutton & Barto, 2018) demonstrates remarkable competencies not only in domains such as chess, video games, and robotics control but also in training foundational models (Ouyang et al., 2022; Wei et al., 2022) and IMIS (Liao et al., 2020; Ma et al., 2021; Li et al., 2021). Given that temporally-extended prompt optimization encompasses both the foundational model and IMIS, we formulate this problem as a Markov decision process (MDP) and employ RL for its resolution. The framework above is then instantiated as the algorithm denoted by **TEPO**. During each stage, TEPO agent determines which prompt form is most suitable for recommendation to human, considering the current segmentation outcomes and historical prompts. The ultimate objective is to augment the performance of SAM in each stage relative to its preceding iteration, thereby maximizing its efficacy.

The contributions encompass three distinct aspects: 1) In an unprecedented discovery, we ascertain that sequential prompt forms constitute the crucial elements influencing the zero-shot performance of SAM in IMIS, subsequently proposing a pertinent temporally-extended prompts optimization problem; 2) By conceptualizing the temporally-extended prompts optimization as an MDP, we employ RL to optimize the sequential selection of prompt forms, thereby enhancing the zero-shot performance of SAM in IMIS; 3) The performance juxtaposition and ablation studies conducted on the standardized benchmark `Brats2020` (Menze et al., 2014) substantiate the efficacy of the TEPO agent in ameliorating SAM's zero-shot capability.

## 2. Related Work and Preliminaries

### 2.1. Interactive Medical Image Segmentation

Before remarkable advancements in automatic segmentation through convolutional neural networks (CNNs), many traditional interactive techniques are employed within IMIS (Zhao & Xie, 2013). Within this scope, the RandomWalk method (Grady, 2006) generates a weight map with pixels as vertices and segments images based on user interaction. Approaches such as GrabCut (Rother et al., 2004) and GraphCut (Boykov & Jolly, 2001) connect image segmentation to graph theory's maximum flow and minimum cut algorithms. Geos (Criminisi et al., 2008) introduces a geodesic distance measurement to ascertain pixel similarity.

Deep learning-based IMIS methods have become a topic of great interest in recent years. Xu et al. (2016) suggests employing CNNs for interactive image segmentation. DeepCut (Rajchl et al., 2016) and ScribbleSup (Lin et al., 2016) utilize weak supervision in developing interactive segmentation techniques. DeepIGeoS (Wang et al., 2018) incorporates a geodesic distance metric to generate a hint map.

Viewing the interactive segmentation process as a sequential procedure lends itself naturally to using reinforcement learning (RL). Polygon-RNN (Castrejon et al., 2017) fundamentally segments target as polygons, iteratively selecting polygon vertices through a recurrent neural network (RNN). While Polygon-RNN+ (Acuna et al., 2018) adopts a similar approach to Polygon-RNN, it employs RL to learn vertex selection. SeedNet (Song et al., 2018) constructs an expert interaction generation RL model capable of obtaining simulated interaction data at each interaction stage.

IteR-MRL (Liao et al., 2020) and BS-IRIS (Ma et al., 2021) conceptualize the dynamic interaction process as a Markov Decision Process (MDP) and apply multi-agent RL models for image segmentation purposes. MECCA (Li et al., 2021) establishes a confidence network based on IteR-MRL, seeking to mitigate the pervasive "interactive misunderstanding" issue that plagues RL-based IMIS techniques and ensure the effective utilization of human feedback. Additionally, Liu et al. (2023) integrates SAM within the *3D Slicer* software, thereby facilitating the process of designing, evaluating, and employing SAM in the context of IMIS.

### 2.2. Segment Anything Model

The *Segmentation Anything Model* (SAM) (Kirillov et al., 2023), recently introduced by Meta, serves as a fundamental framework for tackling image segmentation challenges. Motivated by the robust performance of foundational models in NLP and CV domains, researchers endeavored to establish a unified model for complete image segmentation tasks. Nonetheless, the actual data in the segmentation field necessitates revision and diverges from the design intentions mentioned above. Consequently, Kirillov et al. (2023) stratifies the process into three distinct phases: *task*, *model*, and *data*. Refer to the primary publication (Kirillov et al., 2023) and a contemporary survey (Zhang et al., 2023) for comprehensive explanations.

**Task.** Drawing inspiration from foundational NLP and CV models, Kirillov et al. (2023) introduces the promptable segmentation task to generate a valid segmentation mask in response to any given segmentation prompt. Such prompts delineate the segmentation target within an image and may comprise a location (or point), a range (or bounding box), or a textual description of the object to be segmented. A valid output mask necessitates the production of a plausible mask for at least one target object, even when the prompt is inherently ambiguous or alludes to multiple objects.

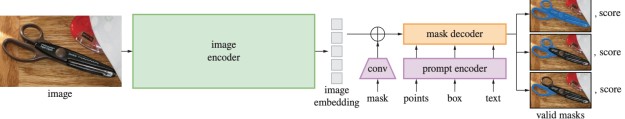

Figure 1: The screenshot of SAM (Kirillov et al., 2023).

**Model.** The promptable segmentation task, paired with the objective of real-world applicability, imposes restrictions on the model architecture. Kirillov et al. (2023) devises a streamlined yet efficacious model, known as SAM (Figure 1), which encompasses a powerful image encoder that computes image embeddings, a prompt encoder that embeds prompts, and a lightweight mask decoder that amalgamates the two information sources to predict segmentation masks.

**Data.** SAM necessitates training on an extensive and diverse collection of masks to attain exceptional generalization capabilities on novel data distributions. Kirillov et al. (2023) constructs a "data engine", employing a model-in-the-loop dataset annotation approach, thereby co-developing SAM in tandem. The resulting dataset, SA-1B, incorporates over 1 billion masks derived from 11 million licensed and privacy-preserving images.

### 2.3. Segment Anything in Medical Images

Building upon the foundational pre-trained models of SAM, many papers have delved into investigating its efficacy in diverse zero-shot MIS scenarios. Ji et al. (2023a) conducts a comprehensive evaluation of SAM in the *everything* mode for segmenting lesion regions within an array of anatomical structures (e.g., brain, lung, and liver) and imaging modalities (*computerized tomography*, abbreviated as CT, and *magnetic resonance imaging*, abbreviated as MRI). Ji et al. (2023b) subsequently scrutinizes SAM's performance in specific healthcare domains (optical disc and cup, polyp, and skin lesion segmentation) utilizing both the automatic *everything* mode and the manual *prompt* mode, employing points and bounding boxes as prompts.

In brain extraction tasks with MRI, Mohapatra et al. (2023) contrasts SAM's efficacy with the renowned *Brain Extraction Tool* (BET), a component of the *FMRIB Software Library*. Deng et al. (2023) appraises SAM's performance in

digital pathology segmentation tasks, encompassing tumor, non-tumor tissue, and cell nuclei segmentation on high-resolution whole-slide imaging. Zhou et al. (2023) adeptly implements SAM in polyp segmentation tasks, utilizing 5 benchmark datasets under the *everything* setting. Recently, an assortment of studies has rigorously tested SAM on over 10 publicly available MIS datasets or tasks (He et al., 2023; Mazurowski et al., 2023; Ma & Wang, 2023; Wu et al., 2023; Huang et al., 2023; Zhang & Liu, 2023).

Quantitative experimental results gleaned from these works reveal that the zero-shot performance of SAM is, on the whole, moderate and exhibits variability across distinct datasets and cases. To elaborate: 1) Utilizing *prompt* instead of *everything* mode, SAM can surpass state-of-the-art (SOTA) performance in tasks characterized by voluminous objects, smaller quantities, and well-defined boundaries when reliant on dense human feedback; 2) However, a considerable performance discrepancy remains between SAM and SOTA methods in tasks involving dense and amorphous object segmentation.

## 3. Method

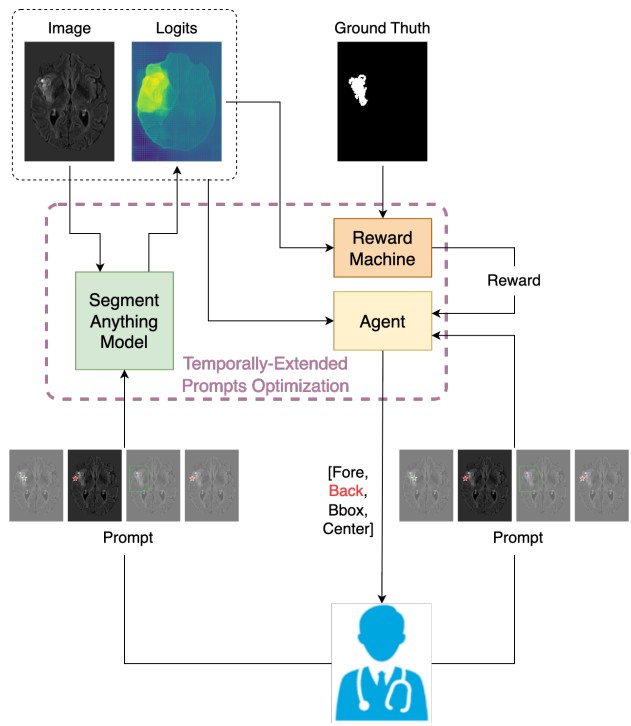

Figure 2: The architecture of our proposed TEPO.

As elucidated in the preceding analysis, the susceptibility of SAM to prompt forms is markedly pronounced in IMIS. This serves as the impetus for devising a mechanism adept at adaptively proffering suitable prompt forms for human specialists, contingent upon the current progression of seg-

mentation. The human expert subsequently imparts feedback to SAM, employing the recommended prompt form. The ensuing discourse delineates the modeling of this mechanism, the temporally-extended prompts optimization, as an MDP (Section 3.1) and elaborates on its resolution through reinforcement learning (Section 3.2).

## 3.1. Problem Formulation

We consider a standard setup consisting of an agent interacting with an environment in discrete finite timesteps. In our setting, the purpose of the agent is to recommend suitable prompt forms for human experts. At each timestep $t$ the agent receives an observation $o_t$, takes an action $a_t$ and receives a scalar reward $r_t$. In general, the environment may be partially observed so that the entire history of the observation, action pairs $s_t = (o_1, a_1, \ldots, a_{t-1}, o_t)$ may be required to describe the state.

An agent's behavior is defined by a policy, $\pi$, which maps states to a probability distribution over the actions $\pi : \mathcal{S} \to \mathcal{P}(\mathcal{A})$. We model it as a Markov decision process with a state space $\mathcal{S}$, action space $\mathcal{A}$, an initial state distribution $p(s_1)$, transition dynamics $p(s_{t+1} \mid s_t, a_t)$, reward function $r(s_t, a_t)$, and instantiate it as follows:

**State space.** The state at timestep $t$ can be represented as a three-tuple $S_t = (I_t, P_{t-1}, T_t)$, where $I_t \in \mathbb{R}^{H \times W \times C}$ is the medical image slice input at time step $t$, $P_{t-1} \in \mathbb{R}^{H \times W \times K}$ represents the segmentation logits from the previous time step $t-1$ (where $K$ represents the number of segmentation classes, which in this case is 2), and $T_t$ is a set of interaction prompts provided before time step $t$. We consider three types of interaction prompts at each timestep: forehead point with label 1, background point with label 0, and bounding box.

**Action space.** The action space $\mathcal{A}$ is a set of interactive forms provided by human experts at each time step. It is represented as a set of integers $\mathcal{A} = \{0, 1, 2, 3\}$, where 0 denotes selecting the forehead point, 1 denotes accessing the background point, 2 denotes the center point, which is defined as the point farthest from the boundary of the error regions, and 3 denotes selecting the bounding box. At each time step, the agent chooses an action from the action space $\mathcal{A}$ to assist human experts with their interactions with SAM.

**Reward function.** At each step t, the difference between the current DICE score (Dice, 1945), $dice(P_t, Y)$ and the previous DICE score, $dice(P_{t-1}, Y)$, is calculated as the reward value $R_t$:

$$R_t = dice(P_t, Y) - dice(P_{t-1}, Y),$$

where $Y$ is the ground truth, $dice(P_t, Y)$ represents the DICE score between the current predicted result $P_t$ and the ground truth, and $dice(P_{t-1}, Y)$ represents the DICE score

between the previous predicted result and the ground truth.

In summary, as shown in Figure 2, the whole process is as follows, the intelligence based on the image, the current segmentation probability and the prompt given by the doctor, according to the strategy pi gives the recommended prompt form, and then the doctor gives the corresponding prompt to SAM according to this, updates the style probability, and the change of the segmentation result forms the reward feedback to the intelligence.

In addition, the *return* from a state is defined as the discounted cumulative reward $\mathcal{R}_t = \sum_{i=t}^{T} \gamma^{(i-t)} r(s_i, a_i)$ with a discounting factor $\gamma \in [0, 1]$. Temporally-extended prompts optimization is then aim to learn a policy that maximizes the expected return from the start distribution $J = \mathbb{E}_{r_i, s_i \sim E, a_i \sim \pi}[\mathcal{R}_1]$. We denote the discounted state visitation distribution for a policy $\pi$ as $\rho^\pi$.

## 3.2. Learning the TEPO Agent with RL

Before introducing specific RL algorithms to obtain the optimal temporarily-extended prompt, we first introduce some notations. The *action-value function* is used in many RL algorithms. It describes the expected return after taking an action $a_t$ in state $s_t$ and thereafter following policy $\pi$ :

$$Q^\pi(s_t, a_t) = \mathbb{E}_{r_{i \geq t}, s_{i > t} \sim E, a_{i > t} \sim \pi}[\mathcal{R}_t \mid s_t, a_t], \quad (1)$$

where $E$ denotes the environment which determines the reward and the next state, and $\mathcal{R}_t$ is the `return` term.

Additionally, many approaches in RL make use of the recursive relationship known as the *Bellman equation*:

$$Q^\pi(s_t, a_t) = \mathbb{E}_{r_t, s_{t+1} \sim E, a_{t+1} \sim \pi}\big[r(s_t, a_t) + \gamma Q^\pi(s_{t+1}, a_{t+1})\big] \quad (2)$$

In this paper, we adopt deep $Q$-network (DQN) (Mnih et al., 2013; 2015) to instantiate the RL framework and learn the TEPO agent. $Q$-learning (Watkins & Dayan, 1992), as the core module of DQN, is a commonly-used, off-policy RL algorithm, uses the greedy policy $\mu(s) = \arg\max_a Q(s, a)$. DQN adapts the $Q$-learning in order to make effective use of large neural networks as action-value function approximators. In order to scale $Q$-learning, DQN introduces two major changes: the use of a replay buffer, and a separate target network for calculating $y_t$. In off-policy reinforcement learning, a replay buffer is a data structure used to store experiences encountered by an agent during its interactions with the environment. By storing past experiences, DQN can mitigate the issues of non-stationarity and correlation of consecutive samples, improving the convergence and robustness of the learned policies.

During the training process, off-policy algorithm DQN randomly samples batches of experiences from the replay

buffer, which are then used to update the agent's policy and value functions. By storing experiences in a replay buffer, off-policy methods can more efficiently use each experience for multiple updates, reducing the variance in the updates and improving sample efficiency.

In this paper, we consider function approximators parameterized by $\theta^Q$, which we optimize by minimizing the loss:

$$\mathcal{L}\left(\theta^Q\right) = \mathbb{E}_{s_t \sim \rho^\beta, a_t \sim \beta, r_t \sim E}\left[\left(Q\left(s_t, a_t \mid \theta^Q\right) - y_t\right)^2\right],$$
(3)

where

$$y_t = r\left(s_t, a_t\right) + \gamma \hat{Q}\left(s_{t+1}, \mu\left(s_{t+1}\right) \mid \theta^Q\right),$$

and $\hat{Q}$ is the target network. The target network's parameters are not trained, but they are periodically synchronized with the parameters of the main Q-network.

# 4. Experiment

This section provides an evaluation of the proposed TEPO on the `BraTS2020` benchmark, which is a prevalent dataset used for MIS tasks. We aim to address the following key questions, and the following evaluation will focus on answering these questions comprehensively, i.e.,

a) Does SAM with multi-step interaction outperform SAM with single-step interaction?

b) Can the policies learned by the TEPO algorithm outperform the rule-based policies?

c) What strategies can be learned from TEPO?

d) How stable are the strategies learned by TEPO?

## 4.1. Dataset and Training Details

SAM requires 2D images as input and 3D images are conventionally often annotated by viewing them in slices, we adopt the practice of slicing the 3D magnetic resonance scans into axial slices, a method commonly used in related research efforts (Wolleb et al., 2022).

To evaluate the effectiveness of TEPO in the context of multi-step interaction, we carefully selected slices with sufficiently large foregrounds in the image. Specifically, we segment the *Whole Tumor* (WT) from the *FLAIR* images and choose slices that contain a minimum of 256 foreground pixel points for analysis. This carefully curated dataset enables accurate evaluation of the performance and potential of TEPO in future applications in MIS.

The dataset for evaluation comprised a total of 369 patients. We split the dataset into three subsets: the training set evaluated 319 patients and included $17,396$ slices; the validation set consisted of 20 patients, corresponding to $1,450$ slices; and the test set included 20 patients with $1,389$ slices.

We crop the images to $200 \times 150$, implement *random flip*, *rotate*, *add noise*, *affine transform* data augmentation to the training dataset, and then rescale the intensity values. We train for 100 epochs, and in each epoch, $10,000$ steps are sampled, and the $Q$ network of TEPO is updated 100 times. The model is trained with a learning rate of $1e^{-3}$ for the `Adam` optimizer and a batch size of 64.

## 4.2. Main Results

The performance of the proposed TEPO algorithm is evaluated on the `BraTS2020` dataset for medical image segmentation tasks and compared with three rule-based policy baselines: the one-step Oracle agent, the random agent, and the alternately changing agent. The one-step Oracle agent is an optimal decision-making agent that has access to comprehensive information and can observe the reward after adapting various interaction forms. This allows it to achieve the highest accuracy in a single step and to explore efficient interaction strategies for the given task. The random agent, on the other hand, uniformly samples actions from available action sets and can be used to simulate clinicians without any preference for any particular interaction form for the task at hand. The alternately changing agent applies a policy that alternately chooses the forehead point and the background point.

We conduct training on several different reinforcement learning policies with varying horizon settings ($N = \{2, 3, 5, 7, 9\}$) and evaluate the agents' performance through the dice score, computed using a ground truth mask and measurements. At each timestep, the agent first chooses an action to indicate what form of interaction is required. To simulate a clinician's behavior, we use rules consisting of choosing specific positions, such as the forehead, background, and center, and drawing bounding boxes around the forehead region. Specifically, we select the forehead, background, and center points that are farthest from the boundaries of the false negative, false positive, and error regions, respectively. For the bounding box, we extend the forehead region by 10 pixels and draw a rectangle.

The comparison of the performance of various interaction strategies is evaluated with respect to the number of interactions. As shown in Figure 3, the different lines correspond to the different agents' performance. "TEPO-X" indicates that the agent is trained in the $X$-step interaction scenario. For example, "TEPO-2" means a two-step scenario. "Random" denotes the random agent, "Alternately" denotes the alternately changing agent, and "1-step Oracle" denotes the one-step Oracle agent. We will use the same labeling convention throughout the paper unless noted otherwise. It is worth noting that we train in different interaction step scenarios, but in testing, we use 9-step interactions to find out comprehensive performances.

Table 1: Action selection preferences and quantitative segmentation performances for TEPO policies. Labels used in the paper include "Fore" for the forehead point form, "Back" for the background point form, "Center" for the center point form, and "Bbox" for the bounding box form. These labels will be consistently used throughout the paper. In addition, The highest dice in each step we indicate with bolded.

| Step | TEPO-2 | | | TEPO-3 | | | TEPO-5 | | | TEPO-7 | | | TEPO-9 | | |
|---|---|---|---|---|---|---|---|---|---|---|---|---|---|---|---|
| | Action | Dice | $< -0.1$ | Action | Dice | $< -0.1$ | Action | Dice | $< -0.1$ | Action | Dice | $< -0.1$ | Action | Dice | $< -0.1$ |
| 1 | Bbox (100.00%) | **0.6901± 0.2094** | 0 | Center (100.00%) | 0.4658± 0.2877 | 0 | Center (100.00%) | 0.4658± 0.2877 | 0 | Center (100.00%) | 0.4658± 0.2877 | 0 | Center (100.00%) | 0.4658± 0.2877 | 0 |
| 2 | Fore (99.57%) | 0.6930± 0.1758 | 95 | Bbox (94.96%) | **0.7035± 0.1882** | 54 | Center (62.35%) | 0.6472± 0.2316 | 117 | Bbox (85.67%) | 0.6981± 0.1965 | 56 | Center (100.00%) | 0.6211± 0.2535 | 177 |
| 3 | Fore (99.78%) | 0.6937± 0.1694 | 14 | Center (98.85%) | **0.7611± 0.1687** | 44 | Center (86.54%) | 0.7369± 0.1926 | 102 | Center (79.34%) | 0.7552± 0.1720 | 56 | Center (100.00%) | 0.7192± 0.2131 | 153 |
| 4 | Fore (99.71%) | 0.6932± 0.1692 | 2 | Center (99.14%) | **0.7845± 0.1670** | 72 | Center (95.10%) | 0.7782± 0.1665 | 103 | Center (90.78%) | 0.7822± 0.1690 | 65 | Center (100.00%) | 0.7707± 0.1711 | 167 |
| 5 | Fore (99.93%) | 0.6940± 0.1693 | 0 | Center (99.78%) | **0.8026± 0.1553** | 73 | Center (97.48%) | 0.8021± 0.1577 | 85 | Center (93.38%) | 0.7991± 0.1612 | 62 | Center (100.00%) | 0.7990± 0.1583 | 123 |
| 6 | Fore (99.86%) | 0.6940± 0.1693 | 0 | Center (100.00%) | **0.8198± 0.1441** | 46 | Center (99.57%) | 0.8190± 0.1439 | 62 | Center (94.89%) | 0.8137± 0.1520 | 53 | Center (100.00%) | 0.8175± 0.1452 | 86 |
| 7 | Fore (99.86%) | 0.6940± 0.1694 | 0 | Center (99.86%) | 0.8263± 0.1409 | 47 | Center (99.64%) | 0.8288± 0.1375 | 48 | Center (95.39%) | 0.8240± 0.1424 | 40 | Center (100.00%) | **0.8302± 0.1390** | 58 |
| 8 | Fore (99.86%) | 0.6940± 0.1694 | 0 | Center (100.00%) | 0.8332± 0.1367 | 45 | Center (99.71%) | 0.8372± 0.1370 | 44 | Center (95.82%) | 0.8316± 0.1370 | 45 | Center (100.00%) | **0.8394± 0.1324** | 51 |
| 9 | Fore (99.93%) | 0.6940± 0.1694 | 0 | Center (100.00%) | 0.8362± 0.1378 | 44 | Center (99.64%) | 0.8421± 0.1322 | 42 | Center (95.61%) | 0.8342± 0.1380 | 41 | Center (100.00%) | **0.8449± 0.1307** | 51 |

Table 2: Action selection preferences and quantitative segmentation performances for rule-based policies.

| Step | Random | | | Alternately | | |
|---|---|---|---|---|---|---|
| | Action | Dice | $< -0.1$ | Action | Dice | $< -0.1$ |
| 1 | Center (26.78%) | 0.4129 ± 0.3417 | 0 | Fore (100.00%) | 0.4658 ± 0.2877 | 0 |
| 2 | Center (29.30%) | 0.5723 ± 0.2947 | 121 | Back (100.00%) | 0.6010 ± 0.2691 | 98 |
| 3 | Fore (30.67%) | 0.6561 ± 0.2562 | 141 | Fore (100.00%) | 0.6460 ± 0.2470 | 207 |
| 4 | Fore (33.48%) | 0.7072 ± 0.2260 | 123 | Back (100.00%) | 0.7280 ± 0.2067 | 98 |
| 5 | Back (32.04%) | 0.7354 ± 0.2094 | 115 | Fore (100.00%) | 0.7332 ± 0.2098 | 172 |
| 6 | Fore (33.48%) | 0.7571 ± 0.1943 | 89 | Back (100.00%) | 0.7823 ± 0.1730 | 59 |
| 7 | Center (33.33%) | 0.7818 ± 0.1727 | 66 | Fore (100.00%) | 0.7840 ± 0.1777 | 111 |
| 8 | Back (33.33%) | 0.7956 ± 0.1627 | 50 | Back (100.00%) | 0.8138 ± 0.1512 | 27 |
| 9 | Fore (33.55%) | 0.8052 ± 0.1568 | 39 | Fore (100.00%) | 0.8052 ± 0.1652 | 92 |

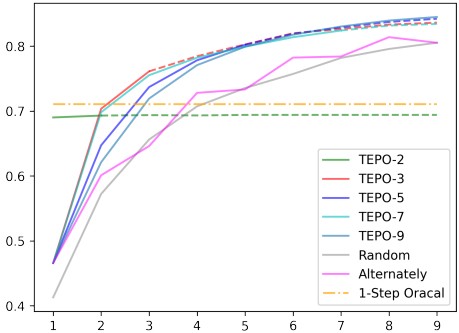

Figure 3: The performance improvement of different interactive medical segmentation methods at different interaction steps. All these test results are performed on the `BraTS2020` dataset.

### 4.2.1. QUANTITATIVE EXPERIMENTAL ANALYSIS

**Q#a: Does SAM in multi-step interaction mode outperform SAM in single-step interaction mode?** As illustrated in Figure 3, the TEPO-2 agent stays the same after the third round, this is because in our experiments, if the shortest distance of all points from the edge in the corresponding region is less than two pixels, then the user does not interact anymore. Table 1 indicates that the TEPO-2 policy predominantly selects the forehead point starting from step two. However, the false negative region is too small to click, so the TEPO-2 policy stops interacting at step five for all test cases. Conversely, the performance of other multi-step policies improves with an increase in the number of interactions, showcasing that SAM can be enhanced through multiple rounds of interactions. Moreover, expect TEPO-2, other policies perform better than the one-step Oracle agent, implying that multi-step interactions are more effective for MIS than the single-step interaction mode.

**Q#b: Can the policies learned by the TEPO algorithm outperform the rule-based policies?** The experimental results in Figure 3 indicate that the TEPO-2 policy performs better than random and alternating selection methods during the initial two interactions. Moreover, the performance of all other RL-based policies is superior to rule-based approaches. These findings provide evidence that the TEPO algorithm significantly boosts the efficacy of SAM in interactive medical scenarios, even in zero-shot mode.

**Q#c: What strategies can be learned from the TEPO algorithm?** As the TEPO algorithm is trained under different interaction round scenarios, the learned strategies exhibit variations, as summarized in Table 1. TEPO-2 employs a straightforward strategy: selecting the bounding box in the first step and the forehead point in subsequent ones. This strategy performs well in the initial two steps, with the performance in the first step nearing that of the one-step Oracle agent that adopts an ideal strategy. TEPO-3 applies a nearly deterministic strategy that chooses the bounding box at the second step and chooses center points at other steps. Moreover, TEPO-5 and TEPO-7 use more uncertain strategies that primarily employ the center point but may resort to alternative ones in the second and third steps. TEPO-9 finds a trivial strategy of choosing the center point at each step, resulting in the best performance in multiple interactions.

**Q#d: How stable are the strategies learned by TEPO?** One issue that may affect the performance of TEPO is interactive misunderstanding, where user interactions result in reduced segmentation dice scores. In this study, we consider interactive misunderstandings when the segmentation dice score decreases by over $0.1$. We analyze the occurrence of interactive misunderstandings for different strategies on our test data, as presented in Table 1 and Table 2. For a more intuitive comparison, we plot the number of interactive misunderstandings for each strategy at different interaction

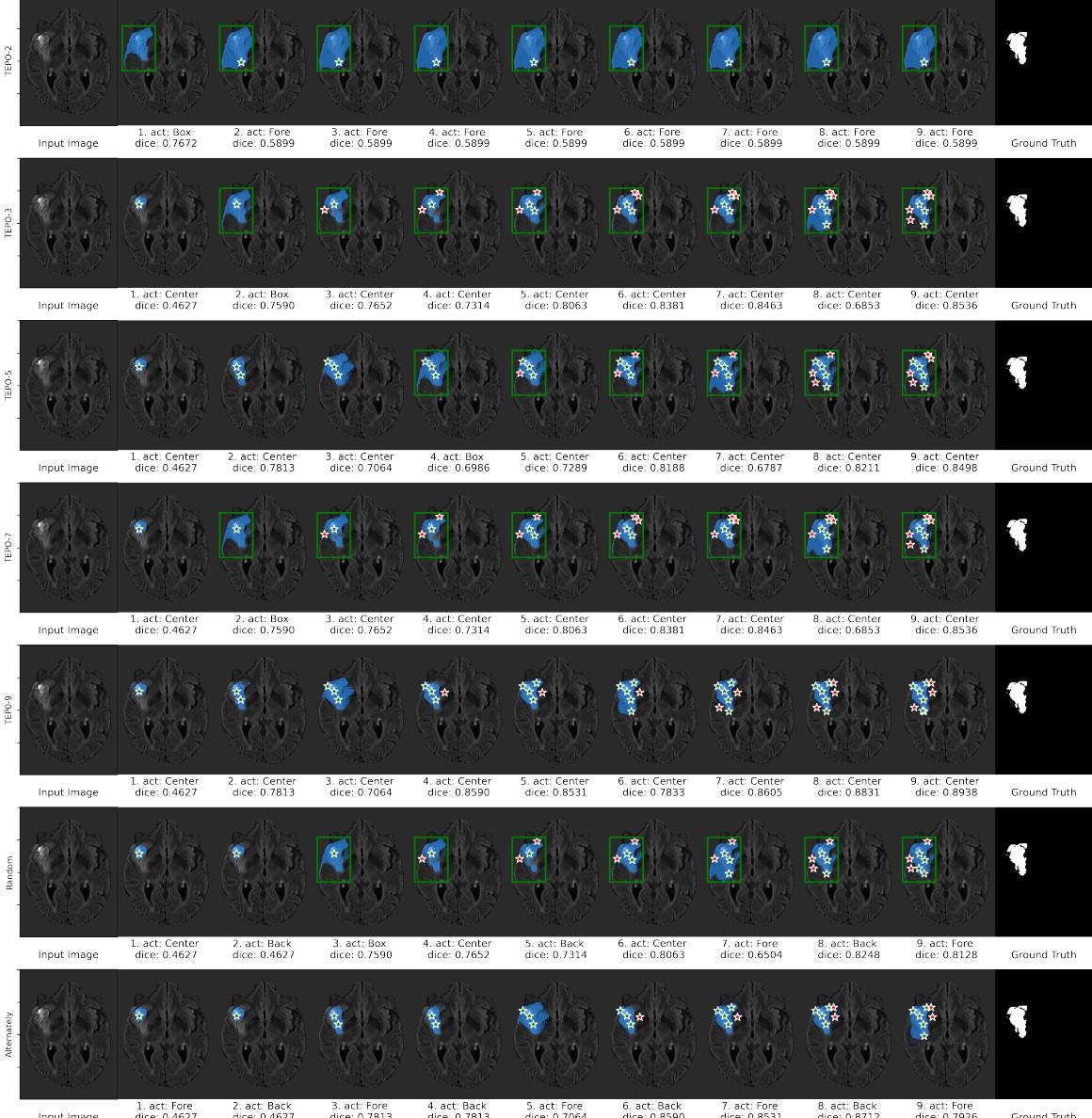

Figure 4: Visualization of strategies and results of different strategies on the same medical image. The green pentagram indicates the foreground, the red pentagram indicates the background, and the green box indicates the bounding box.

steps in Figure 5. As TEPO-2 only applies to the initial two interactions, we exclude it from the plot. Resultantly, the findings indicate that TEPO-3, TEPO-5, and TEPO-7 exhibit superior stability and performance due to fewer misunderstandings than the random and alternating agents.

### 4.2.2. QUALITATIVE EXPERIMENTAL ANALYSIS

To investigate the effectiveness of different strategies, we conducted a qualitative analysis and visualized their performance on a single medical image in Fig.4. The first column shows the raw image, the middle columns show the interaction processes and the segmentation results, and the last column displays the ground truth. Among them, the effective interaction of the TEPO-2 is only twice, so the final style effect is relatively poor. Among other strategies, only the TEPO-9 and alternately changing agent used point-based interaction, reducing the relative efficiency of the error area, but TEPO-9 has an equally good final effect as the strategy with the bounding box. TEPO-3, TEPO-5, and TEPO-7 adopted consistent strategies, where the bounding box was selected in the second interaction and center points were selected in other interactions.

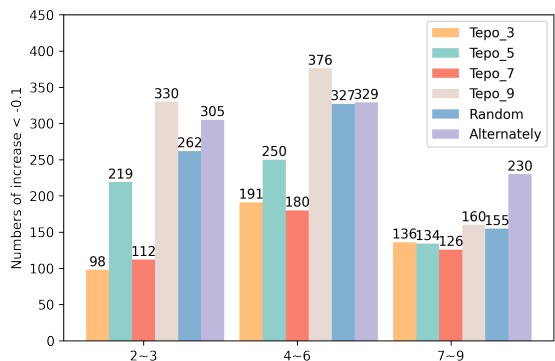

Figure 5: Comparison of the number of cases with increased dice score less than $-0.1$.

Overall, our results reveal that SAM cannot accurately achieve segmentation in a single interaction without being properly tuned. With multiple rounds of interaction, it can achieve considerable results. Moreover, the strategies learned by TEPO demonstrate better segmentation performance compared to rule-based strategies.

## 5. Conclusion

This paper focuses on assessing the potential of SAM's zero-shot capabilities within the interactive medical image segmentation (IMIS) paradigm to amplify its benefits in the medical image segmentation (MIS) domain. We propose an RL mechanism named *temporally-extended prompts optimization* (TEPO) that learns to provide a prompt form that maximizes segmentation accuracy in multi-step interaction scenarios. The experimental results on the `Brats2020` benchmark demonstrate that SAM is prompt-sensitive, and the TEPO agent can further improve its zero-shot capability in the MIS context. The study shows that the proposed TEPO can significantly reduce the number of interactive misunderstandings, which means more robust and stable in providing accurate segmentation in medical images. These findings make a valuable contribution to the development of advanced MIS techniques, showcasing the potential efficacy of prompts optimization which expands the zero-shot capability of foundation models like SAM.

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
