# OpenReview forum: "Temporally-Extended Prompts Optimization for SAM in Interactive Medical Image Segmentation"
_ICML.cc/2023/Workshop/ILHF — ILHF Workshop ICML 2023_

### Official Review · Reviewer_GPGK · 2023-06-06
**Decent Paper but with Minor Flaws**

**Rating:** 6
**Confidence:** 3

**Review:**

**Advantages**

- The paper is well-written. Despite not being an expert in medical imaging or image segmentation, I was able to understand the paper's objective and subject matter.

- The application of RL to the task is well-formulated, making it suitable for a sequential decision-making problem.


**Disadvantages & Areas for Improvement and Suggestions**

- The authors have termed their framework as "Temporally Extended Prompts Optimization (TEPO)". However, from what I perceive, it primarily involves applying the DQN algorithm to a sequential decision-making task. Although the novelty of this approach is apt for the workshop, I feel there might be a tendency to overstate the significance of TEPO. I suggest toning down the language used to present TEPO, emphasizing instead that the authors have applied deep RL to a critical sequential decision-making task. This could provide a more balanced representation of the work's novelty and significance.

- The language in the paper seems unnecessarily complex. This could distract readers, especially those from non-specialist backgrounds. I recommend that the authors undertake further proofreading to simplify the language used and correct any issues.

- Certain concepts could use additional explanation or description. For a broader audience to better understand your work, ensure the paper is self-contained. For instance, the term "intricate anatomical structures" in the introduction requires clarification.

- The paper does not specify whether the authors are considering an MDP over a finite or infinite horizon. This information is necessary.

- The variable "E" used in Eq. 1 is not clearly defined. Over what exactly are the expectations of the states being taken?

- $R_{t}$, the discounted sum of future rewards, is not defined in Eq. 1.

- In **Section 3.2**, the phrase *many approaches in RL make use of the recursive relationship known as the Bellman equation* should be revised. It should reflect that in current RL research, value functions are primarily learned through the Bellman equation.

- To improve the conciseness and flow of Section 3.2, the authors could directly state that they adopt the Q-learning algorithm, subsequently introducing the DQN algorithm for approximating the Q-function with deep neural networks.

- Eq. 2 could be computed over a single expectation, as opposed to the combination of two currently being used.

- The paper does not clarify what a "replay buffer" is, i.e., the off-policy nature of DQN and the meaning of off-policy. This information is necessary in my opinion.

- The requirement for a separate target network is not explained. While specialists might be aware of this, it could leave a gap for general readers.

- Some references are missing. For instance, $\texttt{BRATS2020}$ (referenced in the last paragraph of the first column on page 2) and SA-1B (mentioned under the "data" paragraph on page 3).

- There are some typographical and grammatical errors. For example, use "SAM" instead of "the SAM", refer to Figure 1 as a depiction of the SAM architecture, not a screenshot, and use "details" instead of "details" (in the first paragraph of Section 4). Also, in LaTeX, use ``data engine'' to avoid closing brackets errors. As stated earlier, I recommend a comprehensive proofreading to address these and any other issues.

---

### Official Review · Reviewer_gZnb · 2023-06-16

**Rating:** 8
**Confidence:** 2

**Review:**

The paper proposes TEPO, temporally-extended prompts optimization, an algorithm that recommends the optimal actions (prompt modality, e.g., points or bounding boxes) for the segmentation anything model (SAM) for interactive medical image segmentation (IMIS). The experiments show that TEPO outperforms one-step assessments or the multi-step naive strategy of randomly choosing the prompt modality.

The proposed TEPO framework is simple (building upon ideas in reinforcement learning) yet, I believe, novel. The experiments show positive results.

The paper is for the most part clear and there are only a couple of typos (e.g., "oracal"). One part that needs more description is around how the clinician's behavior is simulated. An experiment with crowdworkers or experts would be an interesting addition to this paper.

---

### Decision · Program_Chairs · 2023-06-20

Accept